# Correlates of Mild Cognitive Impairment of Community-Dwelling Older Adults in Wuhan, China

**DOI:** 10.3390/ijerph15122705

**Published:** 2018-11-30

**Authors:** Xiaojun Liu, Xiao Yin, Anran Tan, Meikun He, Dongdong Jiang, Yitan Hou, Yuanan Lu, Zongfu Mao

**Affiliations:** 1School of Health Sciences, Wuhan University, 115# Donghu Road, Wuhan 430071, China; xiaojunliu@whu.edu.cn (X.L.); yinxiaowhu@whu.edu.cn (X.Y.); chloetar@whu.edu.cn (A.T.); 2017203050026@whu.edu.cn (M.H.); 2017203050046@whu.edu.cn (D.J.); houyitan@whu.edu.cn (Y.H.); 2Global Health Institute, Wuhan University, 8# South Donghu Road, Wuhan 430072, China; yuanan@hawaii.edu; 3College of Public Administration, Huazhong University of Science and Technology, 1037# Luoyu Road, Wuhan 430074, China; 4Department of Public Health Sciences, University of Hawaii at Mānoa, 1960 East-West Road, Honolulu, HI 96822, USA

**Keywords:** mild cognitive impairment (MCI), older adults, vulnerable populations

## Abstract

Mild cognitive impairment (MCI) is an early stage of Alzheimer’s disease or other forms of dementia that occurs mainly in older adults. The MCI phase could be considered as an observational period for the secondary prevention of dementia. This study aims to assess potential differences in the risk of MCI among different elderly groups in Wuhan, China, and to further identify the most vulnerable populations using logistic regression models. A total of 622 older adults participated in this study, and the prevalence of MCI was 34.1%. We found that individuals aged 80–84 (odds ratio, OR = 1.908, 95% confidence interval, 95% CI 1.026 to 3.549) or above (OR = 2.529, 95% CI 1.249 to 5.122), and those with two chronic diseases (OR = 1.982, 95% CI 1.153 to 3.407) or more (OR = 2.466, 95% CI 1.419 to 4.286) were more likely to be diagnosed with MCI. Those with high school degrees (OR = 0.451, 95% CI 0.230 to 0.883) or above (OR = 0.318, 95% CI 0.129 to 0.783) and those with a family per-capita monthly income of 3001–4500 yuan (OR = 0.320, 95% CI 0.137 to 0.750) or above (OR = 0.335, 95% CI 0.135 to 0.830) were less likely to experience MCI. The results also showed that those aged 80 or above were more likely to present with cognitive decline and/or reduced activities of daily living (ADL) function, with the odds ratios being 1.874 and 3.782, respectively. Individuals with two, or three or more chronic diseases were more likely to experience cognitive decline and/or reduced ADL function, with odds ratios of 2.423 and 2.631, respectively. Increased risk of suffering from either MCI and/or decline in ADL functioning is strongly positively associated with older age, lower educational levels, poorer family economic status, and multiple chronic diseases. Our findings highlight that the local, regional, and even national specific MCI-related health promotion measures and interventions must target these vulnerable populations.

## 1. Introduction

The World Alzheimer Report 2018 estimated that more than 50 million people are living with dementia worldwide, and this number will more than triple to 152 million by 2050 [1]. The total estimated worldwide cost of dementia will rise to two trillion U.S. dollars by 2030 [1]. Mild cognitive impairment (MCI) is considered as an early stage of Alzheimer’s disease or other forms of dementia that mainly occurs in older adults [2,3,4,5]. It does not usually interfere with an individual’s daily life, but there is a higher risk of this condition progressing to dementia compared with age-matched controls [5]. In spite of this, there is still an opportunity to prevent the progression to dementia [3,4]. MCI may occur as a transitional stage between normal aging and dementia, and some effective interventions may reduce the conversion rate [3,4,5]. The American Academy of Neurology’s clinical practice guideline on MCI concluded that 14.4–55.6% of people diagnosed with MCI may return to being neurologically intact [5].

MCI is a major public health problem in low- and middle-income countries like China since it is common in older populations, and its prevalence increases with age [1,3,4]. However, this important issue has not yet received sufficient attention from the Chinese society and has not been studied seriously in China, and to date, there is little scientific literature focusing on this topic [6,7]. Studies on MCI are still carried out in developed countries, especially in European countries, the United States, and Japan [8,9,10]. In fact, it is not clear whether the findings from developed countries fit the Chinese sample due to the differences in lifestyle, eating habits, social and cultural customs, education, and other aspects of the built environment. There were more than 240 million elderly residents aged 60 years and above by the end of 2017, accounting for 17.3% of the total population in China [11]. This number will reach 487 million by 2050, in which adults aged 80 years and older will reach 100 million [11]. China’s aging population and elderly population aged 80 years and older have always ranked first worldwide. A recent study estimated that over 10 million Chinese elderly people suffer from Alzheimer’s disease [12]. According to a meta-analysis based on Chinese studies, the estimated number of older people living with MCI in China ranges from 24 million to 40 million [13]. Therefore, with the aggravating trend of the aging population in China, MCI is becoming a critical public health issue in China and will lead to great burden to its health care system, families, and society in the foreseeable future. 

Hence, major policy implementation and public health planning based on high quality research is needed to alleviate the threats and challenges of the aggravating trend of an aging population in China. However, besides studying how to effectively prevent healthy populations from cognitive impairment, future attention must be addressed to those who already present with cognitive impairment symptoms, and by taking active and effective prevention methods, we can alleviate and prevent those populations from deteriorating and developing dementia. In fact, from the perspective of cost effectiveness, for developing countries with massive aging population like China, targeted screening must be conducted before implementing prevention methods to determine the population with the highest risk of MCI, and subsequently administer the related health interventions and support initiatives towards the target population. 

For these reasons, the present study sought to assess the potential differences in the risk of MCI among different community-dwelling elderly people in Wuhan city, China, and thus to identify the most vulnerable populations. Findings from this study will be valuable for the local government to better understand the current situation of MCI prevalence among community-dwelling older people in Wuhan city, China. More importantly, this line of inquiry can provide clues for further research and baseline information useful for the local, regional, and even national governments in China in their attempt to develop more effective health-prevention programs and health risk communication strategies for the targeted population. Similar assessments may be of use in other countries as well, especially in developing countries like India, and as such, the current study may play a role in informing the governments to accomplish the goal of healthy aging, and to build their elderly-friendly communities and cities in future.

## 2. Materials and Methods

### 2.1. Participants

The present study is a cross-sectional, community-based survey study conducted in four communities in Wuhan, China. According to the design, residents who met the following inclusion criteria were considered as our target respondents: (1) individuals aged 65 years old or above, (2) individuals who had household registration in Wuhan or had lived in the community for more than 6 months, and (3) individuals who gave informed consent and voluntarily participated in the survey. However, not all eligible participants were investigated. Potential participants were excluded if they (1) had been absent in the community during household survey for more than three times or had moved out from the residence, (2) could not conduct normal conversation due to aphasia, deafness, blindness, paraplegia, or other critical body illnesses, (3) had other critical illnesses or were at the end of life, (4) had severe mental disorders or had been diagnosed with dementia, (5) had a history of mental illness or congenital mental retardation, and/or (6) had lost their daily living abilities or had severe daily living disabilities. In order to achieve a higher participation rate, we had at least one community administrator in each community to help us in the investigation.

### 2.2. Sample Size Estimation

In this study, the sample size was estimated by the following formula [14]: $$n=\frac{z_{\alpha /2}^{2}\pi (1-\pi )}{\delta^{2}}$$
where *n* is the required sample size, and *z*^2^*_α_*_/2_ is the normal deviate for two-tailed alternative hypothesis at a level of significance. The level of significance (*α*) is set at 5%, and thus the value of *Z*^2^_0.05/2_ equals 1.96 in the present study. *δ* is the desired level of precision, and we desired a 5% precision. *π* is the estimated proportion of an attribute that is present in the population (the MCI prevalence rate in the current study). A Chinese meta-analysis estimated that the prevalence of MCI was between 9.7% and 16.5% [13]. Therefore, according to the formula, the required sample size was determined to be 135 to 212. There should be adequate power since the actual sample size (622) was far more than the required sample size.

### 2.3. Measures

Given our subjects were the elderly aged 65 or above, and considering different education levels of the respondents, a face-to-face interview style was adopted in this study, which took approximately 15 to 20 min to complete. The questionnaire employed for the present study consisted of the following three parts:

#### 2.3.1. Demographic Information

The questionnaire collected information about participants’ socio-demographic characteristics, including participants’ sex, age, marital status, height, weight, education level, family per-capita monthly income (Chinese yuan), and living pattern (living alone/with spouse/with children/with multiple generations). In the present study, we also asked if the participants were diagnosed with any kind of chronic disease in the last six months.

#### 2.3.2. MCI Measure

The present study employed the Chinese version of the AD8 (questionnaire) to screen the cognitive status of the community-dwelling older people. The AD8 is a commonly used screening tool for dementia and mild cognitive impairment in the community setting [15,16,17]. The AD8 contains eight questions asking the respondent to rate change (yes or no) in memory, problem-solving abilities, orientation, and daily activities. It is a brief and sensitive measure that validly and reliably differentiates between cognitively normal individuals and MCI individuals. Items endorsed as “Yes, a change” are summed to yield the total AD8 score, and it is suggested that using a cut-off score of 2 or greater on the AD8 to predict MCI yielded the most desirable combination of sensitivity and specificity. The Chinese version of the AD8 was introduced and developed by Li Tao et al. in 2012 [18,19]. The reliability and validity of the Chinese version of the AD8 was also confirmed [18,19].

#### 2.3.3. Assessment of Activities of Daily Living (ADL)

ADL was assessed using the Chinese version of the Lawton and Brody’s Activities of Daily Living Scale. The Lawton and Brody’s Activities of Daily Living Scale consists of a Physical Self-Maintenance Scale (PSMS) with six questions and an Instrumental Activities of Daily Living Scale (IADL) with eight questions. All questions utilized a four-point Likert scale and the sum-score ranges from 14–56, with a higher score indicating a lower level of ADL functioning. Sum-score >14 should be considered ADL functioning decline and Sum-score >21 should be considered ADL impaired [20]. Studies have confirmed that the Chinese version of the Lawton and Brody’s Activities of Daily Living Scale is suitable for ADL assessments in community-dwelling older people [21,22].

### 2.4. Statistical Analysis

EpiData version 3.1 (The EpiData Association, Odense, Denmark) was applied to set up the database data and the completed questionnaires were double imported into different files separately by different researchers and was cross-checked to ensure the accuracy of the data. All the data analyses for this study were performed using Statistical software the Statistical Package for the Social Sciences (SPSS) version 22.0 for Windows (SPSS Inc., Chicago, IL, USA). The statistical significance level was set at two-sided *p*-values of less than 5% for all tests. Demographic information of the respondents stratified by cognitive status were initially described, and the differences in cognitive status were examined using Chi-square tests. Binary logistic regression models were established to identify the most vulnerable populations who were at higher risk of MCI or comorbid ADL decline and MCI. The results were presented as an odds ratio (OR) value with a 95% confidence interval (95% CI).

### 2.5. Ethical Statements

This study was approved by the Ethics Committee of the School of Health Science, Wuhan University (Project Identification Code: 20120141110052). Informed consent information was attached to each questionnaire and introduced before the surveys. 

## 3. Results

### 3.1. Demographic Characteristics of the Study Sample

The final sample of this study consisted of 622 community-dwelling older adults, including 85 of 264 male respondents (MCI detection rate 32.3%) and 127 of 358 female respondents (MCI detection rate 35.5%). Overall, the age breakdown was as follows: 17.0% 65–69 years, 26.4% 70–74 years, 22.7% 75–79 years, 20.6% 80–84 and 13.3% 85 or more years. Nearly a third (28.6%) of respondents had no spouse (divorced, separated, widowed, or never married, etc.). Respondents’ height and weight were used to calculate their body mass index (BMI), and the BMI was classified as underweight (BMI < 18.5 kg/m^2^), normal (BMI ≥ 18.5 kg/m^2^ and < 24 kg/m^2^), overweight (BMI ≥ 24 kg/m^2^ and < 28 kg/m^2^), and obese (BMI ≥ 28 kg/m^2^) using the Chinese criteria [23], accounting for 9.5%, 46.9 %, 34.4%, and 9.2% of the total sample size, respectively. About 15.8% of the participants had never been educated and 32.8% had only attended primary school. Participants’ family per-capita monthly income ranged from 1501 to 3000 yuan (59.5%) and 3001 to 4500 yuan (21.4%), accounting for the majority of the sample. Of the 622 participants, 107 respondents (17.2%) lived alone, 297 respondents (47.7%) lived only with their spouse, and 155 respondents (24.9%) lived with multiple generations (their children and grandchildren). Most individuals had at least one chronic noncommunicable disease with 26.5% of the participants reporting two chronic diseases, and 24.4% of the participants reporting at least three chronic diseases.

A total of 212 participants (34.1% of the total sample) presented with MCI. Chi-squared tests observed no significant differences between cognitive status across different gender groups (32.3% for males and 35.5% for females with χ^2^ = 0.727, *p* = 0.394, degrees of freedom, *df* = 1), different marital status groups (32.9% for those that were married and 37.1% for others with a χ^2^ = 0.996, *p* = 0.318, *df* = 1), different BMI groups (42.4% for underweight, 31.8% for normal, 31.3% for overweight, and 47.4% for obese with a χ^2^ = 7.664, *p* = 0.053, *df* = 3), and different living pattern (32.7% for those living alone, 31.3% for those living with a spouse, 39.7% for those living with children, and 38.1% for those living with multiple generations, and the χ^2^ = 3.077, *p* = 0.380, *df* = 3). However, cognitive status was significantly affected by age (χ^2^ = 16.076, *p* = 0.003, *df* = 4), educational level (χ^2^ = 29.726, *p* < 0.001, *df* = 4), family per-capita monthly income (χ^2^ = 21.379, *p* < 0.001, *df* = 3), number of chronic diseases (χ^2^ = 19.494, *p* <0.001, *df* = 3), and ADL functioning (χ^2^ = 106.066, *p* < 0.001, *df* = 2). General demographic characteristics of the study sample stratified by cognitive status are summarized in Table 1.

### 3.2. Predictors of MCI among the Elderly

Table 2 illustrates the odds ratios (ORs) obtained from logistic regression model with the 95% confidence intervals (CIs) and *p*-Values. The results confirmed that the increased risk of developing MCI is strongly positively associated with older age. When compared with the reference group (individuals aged 65–69), those aged 80–84 (OR = 1.908, 95% CI 1.026 to 3.549) and those aged 85 and above (OR = 2.529, 95% CI 1.249 to 5.122) were more likely to present with MCI. However, in terms of educational level, those with a high school degree (OR = 0.451, 95% CI 0.230 to 0.883) and college degree or above (OR = 0.318, 95% CI 0.129 to 0.783) were less likely to experience MCI. Those whose family per-capita monthly incomes were 3001–4500 yuan (OR = 0.320, 95% CI 0.137 to 0.750) or above (OR = 0.335, 95% CI 0.135 to 0.830) were also less likely to experience MCI as compared to those making less than 1500 RMB. Further, for those who had two chronic diseases (OR = 1.982, 95% CI 1.153 to 3.407) or above (OR = 2.466, 95% CI 1.419 to 4.286), a higher risk was discovered with respect to having MCI. Significantly, as for ADL functioning, those who had mild disability (OR = 2.171, 95% CI 1.363 to 3.458) and significant disability (OR = 33.715, 95% CI 12.908 to 88.062) showed greater possibilities in having MCI.

### 3.3. Analysis of Influencing Factors for Combination of MCI and ADL

Given that MCI involves cognitive impairments with minimal impairment in instrumental activities of daily living, the current study also evaluated the ADL in older adults. Furthermore, in this study, older adults with normal cognitive and ADL functions were selected as the “Healthy” group, while older adults with either MCI or decline in ADL functioning alone, and individuals with comorbid MCI and decline in ADL functioning were identified as the “Non-complete health” group. From the perspective of cost effectiveness, we believe that the “Non-complete health” group is the most vulnerable group that should be mainly focused upon and intervened with to protect against the conversion of MCI to dementia. 

The matrix diagram of MCI and ADL is shown in Table 3. There are six different combinations based on the status of ADL and cognitive function of the respondents. Among these, older adults with normal cognitive and ADL functions (Type A) were selected as the “Healthy” group, while older adults with either MCI or decline in ADL functioning alone (Type B, C, D, and E), and individuals with comorbid MCI and decline in ADL functioning (Type F) were identified as the “Non-complete health” group. The majority of respondents (338) were detected as Type A in “Healthy” group, while the total of “Non-complete health” groups (Type B, C, D, E, and F) was 284.

To assess the potential differences in different population subgroups, we conducted further analysis between “Healthy” group and “Non-complete health” groups. The results are shown in Table 4. Compared with those aged 65–69, those aged 80–84 and 85 or above were more likely to belong to the “Non-complete health” groups with the odds ratios of 1.874 and 3.782, respectively. However, while comparing with the older adults with underweight BMI, those who were detected with normal (OR = 0.502, 95% CI 0.262 to 0.961) and overweight BMI (OR = 0.422, 95% CI 0.214 to 0.833) were less likely to present with cognitive decline and/or reduced ADL function. Those who had a better education background had lower odds ratios of being in the “Non-complete health” groups. Additionally, our results also indicated that more older adults with two and three or above chronic diseases belonged in the “Non-complete health” groups, with odds ratios of 2.423 and 2.631.

## 4. Discussion

In the foreseeable future, MCI will become a severe burden to families, communities, and the national health care system. However, problems caused by this public health crisis are currently underrated in China. Given the irreversible process of dementia, the MCI phase is considered to be a functional period to aid in the prevention of dementia [3]. In this community-based study, we found that the prevalence of MCI was 34.1% in Wuhan, China. Our result is higher than that reported in previous studies in China. Lu et al.’s study estimated a MCI prevalence of 26.6% among the Chinese elderly aged 50 to 70 years [24]. Ding et al. revealed a prevalence of more than 20.0% for total MCI in Shanghai, China [25]. Xu et al. reported that the pooled prevalence of MCI for community-dwelling people aged 60–69 years was 28.3% [26]. However, this is understandable due to the fact that the majority of respondents in present studies were over 70 years old (with only 17% under the age of 70). Furthermore, unlike the report conducted in Shanghai by Ding’s group [26], participants in this study generally had a lower educational background, and the family per-capita monthly income of our respondents was relatively lower, with nearly 70% of all the participants having a family per-capita monthly income of less than 3000 yuan (equivalent to 432 U.S. dollars using an exchange rate of 1 U.S. dollar to 6.93 yuan).

Consistent with the conclusions reported by many previous studies, the current study confirmed that the increased risk of developing MCI is strongly positively associated with older age and lower educational levels. Moreover, we found that family economic status was also significantly correlated with the cognitive state of the elderly. Specifically, the elderly who had better family finances were less likely to experience MCI. A community-based epidemiological cohort study also showed that the MCI is significantly linked with poorer financial status and health care decision-making [27]. Therefore, elderly individuals with poor financial status should be considered a subgroup especially vulnerable to MCI. According to our logistic regression analysis, the number of chronic diseases diagnosed was positively correlated with the MCI detection rate. Elderly people with two and three or more chronic diseases have a 1.982 and 2.466 odds of MCI detection rate, respectively, compared to odds for those with no chronic disease, indicating chronic disease as a possible risk factor in developing MCI.

Given that MCI involves cognitive impairments with minimal impairment in instrumental activities of daily living (ADL), considerable empirical studies have widely validated the association between cognitive decline and decline in the ADL function of the elderly [28,29,30,31,32,33]. This result was also confirmed in the current study since we also evaluated the ADL in older adults. Furthermore, in this study, older adults with normal cognitive and ADL functions were selected as “Healthy” group, while older adults with either MCI or decline in ADL functioning alone, and individuals with comorbid MCI and decline in ADL functioning were identified as “Non-complete health” group. From the perspective of cost effectiveness, we believe that the “Non-complete health” group is the most needed vulnerable group that should be mainly focused and intervened to protect against the conversion of MCI to dementia. The present paper revealed that more older adults aged 80–84 and 85 or older were detected in the “Non-complete health” group with the odds of 1.874 and 3.782 as compared to those aged 65–69. It was also found that more older adults with multiple chronic diseases were detected in the “Non-complete health” group. Older adults who received higher education levels were less likely to be detected in the “Non-complete health” group.

One of the major findings from this study is that BMI was unrelated to cognitive status, but significantly linked with the combined ADL/MCI variable. This pattern of findings suggests that BMI is associated with ADL among community-dwelling older adults. Those whose BMI is normal and overweight appear to have smaller odds ratios of having a reduced ADL function as compared with those with underweight BMI, which is consistent with the reports from previous studies. A 22-year cohort study conducted in Japan demonstrated that the relationship between BMI and risk of ADL decline was U-shaped among women, indicating women with normal or overweight BMI have lower risk of ADL decline. The study also stated that men with normal BMI had lower risks of ADL decline [34]. However, Ritchie et al.’s study declared that BMI categories were not associated with ADL decline among community-dwelling older adults [35]. Therefore, association between BMI and ADL decline in older adults remains to be determined.

In China, people generally tend to pay greater attention to the treatment of the disease, and it is easy to ignore the prevention and control of the disease. This is especially true for MCI. The main potential reasons may be that MCI is highly secretive, has little impact on daily life [5], and is not easy to detect [8]. On the other hand, the prevention and control of MCI is of great intensity and high cost, yet the actual effects are difficult to evaluate. Consequently, the dilemma will be getting more and more difficult to resolve. To prevent and control MCI, and to reduce the conversion of MCI into dementia from the perspective of cost-effectiveness, developing countries like China should carry out health interventions aimed at the elderly who are most in need and are most likely to suffer from MCI. Older people with higher age, lower educational levels, and with multi-morbidity of chronic diseases should be considered as those most vulnerable, who are most in need, and who require the most attention and health interventions. In addition, targeted health interventions, public education, and support initiatives should also include those elderly people who are underweight or obese. 

The following limitations of the present study should be noted: firstly, our cross-sectional study design can only understand the current the status of, and factors associated with MCI prevalence among community-dwelling older people in Wuhan city, China. In addition, we did not track how many community-dwelling elderly people in total were contacted, and how many of them declined or were excluded in this study. Therefore, non-response bias was not assessed and as such, it may have affected the results. For future studies, all the limitations stated above should be taken into account and try to avoid. Given the fact that few evidence could support intervention measures can effectively prevent conversion form MCI to dementia, specific health intervention experiments and effectiveness evaluations regarding the prevention and control of MCI on target population should be extensively emphasized in future, and more in-depth studies need to be taken to understand how specific health interventions come into effect [36,37,38].

## 5. Conclusions

In conclusion, the current study has confirmed that older age and lower level of education are two major risk factors strongly associated with the development of MCI. The results of this study also revealed that elderly individuals with poorer family economic status and multiple chronic diseases are especially vulnerable to MCI. In addition to MCI, these mentioned factors are also positively associated with declined function of ADL, or with comorbid MCI and an overall decline in ADL functioning. Older adults with a BMI between 18.5 and 28 are less likely to suffer from either MCI or decline in ADL functioning alone, or comorbid MCI and decline in ADL functioning. Our findings highlight that the local, regional, and even national specific MCI-related health promotion measures, interventions, and/or support initiatives (including medical attention, early diagnosis, and active treatments, etc.) in developing countries like China need to target these most vulnerable populations who are perceived to be worsening in regards to cognitive status and disability and/or are at highest risk of MCI, in order to reduce and control the development of dementia among the older adult population.

## Figures and Tables

**Table 1 ijerph-15-02705-t001:** Demographic information of participants stratified by cognitive status (*n* = 622).

Demographic Information	Total Population	Normal	MCI	χ^2^ (*df*)	*p-*Value
		*n* = 410 (65.9%)	*n* = 212 (34.1%)		
Gender				0.727	0.394
Male	264 (42.4)	179 (67.8)	85 (32.3)	(1)	
Female	358 (57.6)	231 (64.5)	127 (35.5)		
Age (years)				16.076	0.003
65–69	106 (17.0)	77 (72.6)	29 (27.4)	(4)	
70–74	164 (26.4)	121 (73.8)	43 (26.2)		
75–79	141 (22.7)	94 (66.7)	47 (33.3)		
80–84	128 (20.6)	72 (56.3)	56 (43.8)		
≥85	83 (13.3)	46 (55.4)	37 (44.6)		
Marital status				0.996	0.318
Married	444 (71.4)	298 (67.1)	146 (32.9)	(1)	
Others (Divorced, separated, widowed, never married, etc.)	178 (28.6)	112 (62.9)	66 (37.1)		
BMI				7.664	0.053
Underweight	59 (9.5)	34 (57.6)	25 (42.4)	(3)	
Normal	292 (46.9)	199 (68.2)	93 (31.8)		
Overweight	214 (34.4)	147 (68.7)	67 (31.3)		
Obesity	57 (9.2)	30 (56.2)	27 (47.4)		
Educational level				29.726	<0.001
Uneducated	98 (15.8)	46 (46.9)	52 (53.1)	(4)	
Primary school	204 (32.8)	127 (62.3)	77 (37.7)		
Junior middle school	164 (26.4)	114 (69.5)	50 (30.5)		
High school	107 (17.2)	83 (77.6)	24 (22.4)		
≥College	49 (7.9)	40 (81.6)	9 (18.4)		
Family per-capita monthly income (yuan)				21.379	<0.001
≤1500	40 (6.4)	21 (52.5)	19 (47.5)	(3)	
1501–3000	370 (59.5)	224 (60.5)	146 (39.5)		
3001–4500	133 (21.4)	104 (78.2)	29 (21.8)		
>4500	79 (12.7)	61 (77.2)	18 (22.8)		
Living pattern				3.077	0.380
alone	107 (17.2)	72 (67.3)	35 (32.7)	(3)	
with spouse	297 (47.7)	204 (68.7)	93 (31.3)		
with children	63 (10.1)	38 (60.3)	25 (39.7)		
with multiple generations	155 (24.9)	96 (61.9)	59 (38.1)		
Number of chronic diseases				19.494	<0.001
0	133 (21.4)	100 (75.2)	33 (24.8)	(3)	
1	172 (27.7)	126 (73.3)	46 (26.7)		
2	165 (26.5)	101 (61.2)	64 (38.8)		
≥3	152 (24.4)	83(54.6)	69 (45.4)		
ADL functioning				106.066	<0.001
Normal	443 (71.2)	338 (76.3)	105 (23.7)	(2)	
Declined	119 (19.1)	65 (54.6)	54 (45.4)		
Impaired	60 (9.6)	7 (11.7)	53 (88.3)		

Note: *df* = degrees of freedom. ADL: activities of daily living; MCI: mild cognitive impairment; BMI: body mass index.

**Table 2 ijerph-15-02705-t002:** Logistic regression analysis for the demographic factors associated with MCI. CI: confidence interval.

Variables	OR	95% CI		*p-*Value
		Lower	Upper	
Gender				
Female	1.272	0.851	1.900	0.241
Age (years)				
70–74	0.900	0.496	1.633	0.729
75–79	1.301	0.711	2.381	0.394
80–84	1.908	1.026	3.549	0.041
≥85	2.529	1.249	5.122	0.010
Marital status				
Others	1.120	0.592	2.118	0.729
BMI				
Normal	0.688	0.365	1.297	0.248
Overweight	0.679	0.349	1.323	0.256
Obesity	1.179	0.561	2.668	0.692
Education				
Primary school	0.681	0.400	1.160	0.157
Junior middle school	0.590	0.328	1.061	0.078
High school	0.451	0.230	0.883	0.020
≥College	0.318	0.129	0.783	0.013
Family per-capita monthly income (yuan)				
1501–3000	0.732	0.357	1.503	0.395
3001–4500	0.320	0.137	0.750	0.009
>4500	0.335	0.135	0.830	0.018
Living pattern				
with spouse	1.306	0.618	2.761	0.484
with children	1.549	0.763	3.147	0.226
with multiple generations	1.780	0.878	3.609	0.110
Number of chronic diseases				
1	1.070	0.616	1.856	0.810
2	1.982	1.153	3.407	0.013
≥3	2.466	1.419	4.286	0.001
ADL functioning				
Declined	2.171	1.363	3.458	0.001
Impaired	33.715	12.908	88.062	<0.001

Note: OR = odds ratio; CI = confidence interval; BMI = body mass index; ADL = activities of daily living.

**Table 3 ijerph-15-02705-t003:** Combination of MCI and ADL.

	Cognition	Normal	Declined	Impaired
ADL				
Normal		Type A (338)	Type B (65)	Type C (7)
MCI		Type D (105)	Type E (54)	Type F (53)

**Table 4 ijerph-15-02705-t004:** Associations between demographic characteristics and the combination of MCI and ADL.

Variables	OR	95%CI		*p-*Value
		Lower	Upper	
Gender				
Female	1.185	0.805	1.746	0.390
Age (years)				
70–74	0.754	0.432	1.318	0.322
75–79	1.233	0.699	2.177	0.469
80–84	1.874	1.035	3.390	0.038
≥85	3.782	1.865	7.670	0.000
Marital status				
Others	1.136	0.599	2.155	0.696
BMI				
Normal	0.502	0.262	0.961	0.038
Overweight	0.422	0.214	0.833	0.013
Obesity	0.689	0.298	1.596	0.385
Education				
Primary school	0.485	0.276	0.853	0.012
Junior middle school	0.445	0.424	0.816	0.009
High school	0.458	0.237	0.888	0.021
≥College	0.247	0.107	0.584	0.001
Family per-capita monthly income (yuan)				
1501–3000	1.062	0.509	2.214	0.873
3001–4500	0.505	0.218	1.170	0.111
>4500	0.557	0.229	1.357	0.198
Living pattern				
with spouse	1.044	0.500	2.183	0.909
with children	2.225	0.881	4.547	0.208
with multiple generations	1.594	0.789	3.221	0.194
Number of chronic diseases				
1	1.262	0.751	2.120	0.379
2	2.423	1.433	4.098	0.001
≥3	2.631	1.530	4.527	0.000

Note: OR = odds ratio; CI = confidence interval; BMI = body mass index; ADL = activities of daily living.

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
