# Peer review of "Correlates of Mild Cognitive Impairment of Community-Dwelling Older Adults in Wuhan, China"

_ijerph, 2018, doi:10.3390/ijerph15122705_

Round 1

Reviewer 1 Report

The present research examines the prevalence and correlates of mild cognitive impairment (MCI) in a community sample of adults aged 65+ years in Wuhan, China. Prevalence was estimated to be 34.1%. Old age, chronic diseases, low education level, low income, and low daily life functioning were associated with higher MCI prevalence.

The research targets an interesting research question, and the analyses were done competently. The manuscript is overall well written and the findings are presented in a clear and concise manner; I enjoyed reading this work.

Although there is a lot to like about this work there remain some issues that need to be addressed.

1)     Perhaps the most important issue is the question: what have we learned from this study? The findings that MCI is associated with age, education, … is not new and has been shown in previous work. Of course, most of this previous work has targeted MCI in European / US samples (a fact emphasized by the authors), but the authors fail to provide a convincing argument (or any argument, really) why the correlates of MCI should be different (or not) in the sample investigated here.

2)     The authors seem very confident that interventions targeting MCI can be effective in preventing people with MCI from developing dementia. I do not share their optimism. The authors write that MCI “is considered as an early stage of Alzheimer’s disease or other forms of dementia” (l.44). If MCI is an early stage of neurological degeneration, I am not convinced that there are many effective interventions to slow down this degeneration. I think the authors need to provide convincing evidence that interventions are in fact effective in preventing conversion form MCI to dementia if they want to uphold this argument. If such evidence cannot be provided, this argument should be toned down considerably.

3)     Sample size estimation: For what test was the sample size planned? I am not familiar with the formula presented by the authors. Could they maybe elaborate more on the sample size estimation?

4)     Sample representativeness: One of the aims of the present study was to estimate the prevalence of MCI in Wuhan, China. The validity of such an estimate heavily depends on the representativeness of the sample. However, based on the information provided by the authors, it is not possible to judge this criterion. Could the authors maybe elaborate on the representativeness of the sample?

5)     Throughout the manuscript, the authors refer to the logistic regression as “multivariate” (e.g., l .149) or “multivariable” (e.g., l .22). The former is a misnomer, because multivariate refers to a model with several dependent variables. The latter expression seems rather unconventional. I would suggest to simply refer to these models as logistic regression models.    

6)     The purpose of the analyses reported in section 3.3. (combining MCI and ADL) is unclear. To me these analyses felt like an afterthought, and they do not really seem necessary. I would suggest to either (1) remove theses analyses, or (2) set up the reasoning for these analyses upfront in the introduction (if they were planned in an a-priori fashion), or (3) provide the reasoning for the (post-hoc) analyses in the respective section 3.3.

7)     I think that the authors over-interpret their findings regarding BMI: this variable was unrelated to MCI, and only related to the combined MCI/ADL variable investigated in section 3.3. Because the latter analyses seem rather post hoc to me (see my comment 6), the interpretation regarding BMI should be toned down (e.g. in the Conclusion section).

8)     “We believe that the “Non-complete health” group is the most needed vulnerable group that should be mainly focused and intervened to protect against the conversion of MCI to 268 dementia.” (l. 267-269): This statement is trivial: Because only (some of the individuals) in this group actually have MCI, it logically follows that only this group (and not the healthy group) should be targeted by an intervention

9)     “Results from this study argue that combined efforts from individuals, caregivers, families, communities, government organizations, and non-governmental organizations should be directed toward MCI-related education among community-dwelling older adults.” (l.291-293): This conclusion does not follow from the study’s results. While I might agree that MCI-related education could be helpful, this does not follow from the results presented in the current work (which examined prevalence and correlates of MCI).

Minor issues

10) Degrees of freedom for all chi-square values should be reported in brackets (e.g., chi-sq(1) = …)

11) l.59: “There were more than 240 million elderly residents […]”: What age range would the authors consider to represent “elderly”? Is it 70+ years? 80+ years?

12) l.206: “Further, for those had two […]” à Should be “Further, for those WHO had two […].”

Author Response

November 18, 2018

Editor-in-Chief

Int. J. Environ. Res. Public Health

Dear Editors and Reviewers,

We are respectfully submitting a revised version of our manuscript, entitled “Correlates of Mild Cognitive Impairment of Community-Dwelling Older Adults in Wuhan, China, for your consideration for publication in the topical collection "Aging and Public Health" of International Journal of Environmental Research and Public Health.

The authors very much appreciate the thoughtful and critical feedback from the reviewers. We are delighted at your decision. The reviewers are clearly familiar with the topic, as is exemplified by their close and accurate review of this manuscript. The manuscript has been revised according to the reviewers comments, and all changes have been highlighted for ready identification. In addition, we have addressed each of the comments from you specifically, and our responses have been outlined in a comment/response format below (see responses to the comments).

Hope the revision is satisfactory and this manuscript is now acceptable for publication in your journal.

I am looking forward to hearing from you soon.

Sincerely,

Zongfu Mao, Ph.D.

Professor and Director

Global Health Institute

School of Health Science

Wuhan University

115# Donghu Road, Wuhan City 430071,P.R.China

Responses to the reviewers’ comments

Reviewer #1

COMMENTS

The present research examines the prevalence and correlates of mild cognitive impairment (MCI) in a community sample of adults aged 65+ years in Wuhan, China. Prevalence was estimated to be 34.1%. Old age, chronic diseases, low education level, low income, and low daily life functioning were associated with higher MCI prevalence.

The research targets an interesting research question, and the analyses were done competently. The manuscript is overall well written and the findings are presented in a clear and concise manner; I enjoyed reading this work.

Response: We thank the reviewer for this supportive comment.

Comment #1) Perhaps the most important issue is the question: what have we learned from this study? The findings that MCI is associated with age, education, … is not new and has been shown in previous work. Of course, most of this previous work has targeted MCI in European / US samples (a fact emphasized by the authors), but the authors fail to provide a convincing argument (or any argument, really) why the correlates of MCI should be different (or not) in the sample investigated here.

Response: We thank and agree with the reviewer for this thoughtfully comment. 

MCI is a major public health problem in low- and middle-income countries with a rapidly aging population and a large elderly population, like China. However, this important issue has not yet received sufficient attention from the Chinese society and has not been studied seriously in China. A number of related studies has been done in developed countries. However, it is not clear whether the findings from developed countries fit the Chinese sample. That's one of the most important reasons why we conducted this research.

We have responded to this comment by adding the following statement in the revised text: “In fact, it is not clear whether the findings from developed countries fit the Chinese sample due to the differences in lifestyle, eating habits, social and cultural customs, education and the other built environment.

Comment #2) The authors seem very confident that interventions targeting MCI can be effective in preventing people with MCI from developing dementia. I do not share their optimism. The authors write that MCI “is considered as an early stage of Alzheimer’s disease or other forms of dementia” (l.44). If MCI is an early stage of neurological degeneration, I am not convinced that there are many effective interventions to slow down this degeneration. I think the authors need to provide convincing evidence that interventions are in fact effective in preventing conversion form MCI to dementia if they want to uphold this argument. If such evidence cannot be provided, this argument should be toned down considerably.

Response: We appreciate this comment and agree that the relevant argument should be toned down. The American Academy of Neurology's clinical practice guideline on MCI concluded that MCI is considered as an early stage of Alzheimer’s disease or other forms of dementia, yet 14.4%–55.6% of people diagnosed with MCI may return to being neurologically intact. But, we’re not sure that there are many effective measures to slow down this degeneration. So we call on specific health intervention experiments and effectiveness evaluations regarding the prevention and control of MCI on target population should be extensively emphasized in future and more in-depth studies need to be taken to understand how specific health interventions come into effect.

What’s more, we've made the changes accordingly (i.e., MCI MAY occur as a transitional stage between normal aging and dementia, and some effective interventions may reduce the conversion rate). We also have added more detail to the Limitations sections. (the last paragraph of the discussion)

Comment #3) Sample size estimation: For what test was the sample size planned? I am not familiar with the formula presented by the authors. Could they maybe elaborate more on the sample size estimation

Response: The sample size for this study was determined according to the formula for the cross-sectional survey study (Because the design of this study was a cross-sectional survey study). In the formula, n is the required sample size, z2α/2 is the normal deviate for two-tailed alternative hypothesis at a level of significance, the level of significance (α) is set at 5%, thus the value of Z20.05/2 equals 1.96 in the present study. δ is the desired level of precision, and we desired a 5% precision. π is the estimated proportion of an attribute that is present in the population (the MCI prevalence rate in the current study). A Chinese meta-analysis estimated that the prevalence of MCI was between 9.7% and 16.5%. Therefore, according to the formula, the required sample size was determined to be 135 to 212.

Comment #4) Sample representativeness: One of the aims of the present study was to estimate the prevalence of MCI in Wuhan, China. The validity of such an estimate heavily depends on the representativeness of the sample. However, based on the information provided by the authors, it is not possible to judge this criterion. Could the authors maybe elaborate on the representativeness of the sample?

Response: We appreciate this comment and agree that the representativeness of the sample is extremely important. In order to reduce bias as much as possible, we developed and implemented very strict sample inclusion and exclusion criteria. (in the 2.1. Participants section: According to the design, residents who met the following inclusion criteria were considered as our target respondents: 1) 65 years old or above, 2) household registration in Wuhan or those who had lived in the community for more than 6 months, 3) informed consent and voluntarily participated the survey. However, not all eligible participants were investigated. Potential participants were excluded if they 1) had been absent in the community during household survey for more than three times or had moved out from the residence, 2) could not conduct normal conversation due to aphasia, deafness, blindness, paraplegia, or other critical body illnesses, 3) had other critical illnesses or were at the end of life, 4) had severe mental disorders or had been diagnosed with dementia, 5) had a history of mental illness or congenital mental retardation, 6) had lost their daily living abilities or had severe daily living disabilities.). Besides, the actual sample size was far more than the required sample size.

Comment 5) Throughout the manuscript, the authors refer to the logistic regression as “multivariate” (e.g., l .149) or “multivariable” (e.g., l .22). The former is a misnomer, because multivariate refers to a model with several dependent variables. The latter expression seems rather unconventional. I would suggest to simply refer to these models as logistic regression models.

Response: We thank the reviewer for this excellent suggestion and have made the change accordingly through the text.

Comment #6) The purpose of the analyses reported in section 3.3. (combining MCI and ADL) is unclear. To me these analyses felt like an afterthought, and they do not really seem necessary. I would suggest to either (1) remove theses analyses, or (2) set up the reasoning for these analyses upfront in the introduction (if they were planned in an a-priori fashion), or (3) provide the reasoning for the (post-hoc) analyses in the respective section 3.3. 

Response: We have added more context to the text in the section 3.3. Please see the revised section.

Text added to the section 3.3:

“Given that MCI involves cognitive impairments with minimal impairment in instrumental activities of daily living. The current study also evaluated the ADL in older adults. Furthermore, in this study, older adults with normal cognitive and ADL functions were selected as “Healthy” group, while older adults with either MCI or decline in ADL functioning alone, and individuals with comorbid MCI and decline in ADL functioning were identified as “Non-complete health” group. From the perspective of cost effectiveness, we believe that the “Non-complete health” group is the most needed vulnerable group that should be mainly focused and intervened to protect against the conversion of MCI to dementia.”

Comment #7) I think that the authors over-interpret their findings regarding BMI: this variable was unrelated to MCI, and only related to the combined MCI/ADL variable investigated in section 3.3. Because the latter analyses seem rather post hoc to me (see my comment 6), the interpretation regarding BMI should be toned down (e.g. in the Conclusion section).

Response: We thank the reviewer for bringing up this question. We have removed the relevant statement in the Conclusion section.

Comment 8) “We believe that the “Non-complete health” group is the most needed vulnerable group that should be mainly focused and intervened to protect against the conversion of MCI to 268 dementia.” (l. 267-269): This statement is trivial: Because only (some of the individuals) in this group actually have MCI, it logically follows that only this group (and not the healthy group) should be targeted by an intervention.

Response: We wrote in the introduction: “In fact, from the perspective of cost effectiveness, for developing countries with massive aging population like China, targeted screening must be conducted before implementing prevention methods to determine the population with the highest risk of MCI, and therefore, conduct the related health interventions and support initiatives towards the targeted population.” That’s why we believe that the “Non-complete health” group is the most needed vulnerable group that should be mainly focused and intervened (but only this group).

We have added “From the perspective of cost effectiveness” as a prerequisite for this statement.

Comment 9) “Results from this study argue that combined efforts from individuals, caregivers, families, communities, government organizations, and non-governmental organizations should be directed toward MCI-related education among community-dwelling older adults.” (l.291-293): This conclusion does not follow from the study’s results. While I might agree that MCI-related education could be helpful, this does not follow from the results presented in the current work (which examined prevalence and correlates of MCI).

Response: We've removed these statements.

Comment 10) Degrees of freedom for all chi-square values should be reported in brackets (e.g., chi-sq(1) = …)

Response: We have made this revision.

Comment 11) l.59: “There were more than 240 million elderly residents […]”: What age range would the authors consider to represent “elderly”? Is it 70+ years? 80+ years?

Response: We thank the reviewer for pointing out this deficiency and we have revised manuscript and made it more clear now.

Comment 12) l.206: “Further, for those had two […]” à Should be “Further, for those WHO had two […].”

Response: We thank the reviewer for pointing this error out and we have revised the statement.

Again, we thank the reviewer for all these valuable review comments.

Reviewer 2 Report

Summary

This paper reports the results of a cross-sectional study on different demographic, health and socioeconomic factors related to MCI among community-dwelling older people in Wuhan, China. The sample size is adequate. Overall, the paper is well-written and the study is scientifically sound. The importance of the study is well founded in the introduction which also gives adequate background information on the significance of measuring MCI. The methods are mostly well-reported. The results are reported thoroughly and precisely. The discussion section summarizes the findings and relates them to previous knowledge on the subject. Moreover, the significance of measuring MCI and the implications of the current results are well-discussed. 

I feel the paper adds relecant knowledge to the literature on MCI and related factors, especially given the specific context of China. However, I have some comments that need to be addressed before the paper could be published. 

Broad comments: 

Title – the title needs to be revised, as it is now it gives the impression that the study is a methods study on measuring MCI. So please revise as to better describe the actual content of the study (i.e. correlates of MCI, or factors related to MCI etc.)

The participant flow needs to be described more thoroughly, e.g. how was the study population recruited, and through which phases was the total of 622 persons reached (e.g. how many were contacted, how many of them declined or were excluded from the study based on the different criteria, how did the sample included in the study differ from those who were excluded or did not consent to the study). I would also like to know how the face-to-face interviews were organized. Furthermore, I would like to know how the authors justify the use of a much bigger sample than needed based on the sample size calculations|. 

I wonder did the authors consider conducting also adjusted logistic regression models to examine which factors maybe explained the current associations. It would be interesting to know which of the factors had the strongest association with MCI despite other contributing factors. 

The discussion sections lacks strengths and limitations of the current study, please add these, especially the limitations need to be added, e.g.on the selection of the study sample. 

Specific comments: 

The first sentence of the results section is unclearly formulated, please correct to be more clear. This may be a language issue, I would recommend language editing throughout the manuscript as I noticed some other minor language issues as well. 

On page 5, line 189 “were less likely” is not correctly phrased based on a chi-square test, please state a fifferene and not likelihood. 

The result tables are quite large and therefore I would omit reporting betas, standard deviations and wald tests in the tables, these are not needed when reporting odds ratios, confidense limits and p-values. 

Table 3 was not clear to me, please clarify in the table captions what the types and numbers refer to. 

Author Response

November 18, 2018

Editor-in-Chief

Int. J. Environ. Res. Public Health

Dear Editors and Reviewers,

We are respectfully submitting a revised version of our manuscript, entitled “Correlates of Mild Cognitive Impairment of Community-Dwelling Older Adults in Wuhan, China, for your consideration for publication in the topical collection "Aging and Public Health" of International Journal of Environmental Research and Public Health.

The authors very much appreciate the thoughtful and critical feedback from the reviewers. We are delighted at your decision. The reviewers are clearly familiar with the topic, as is exemplified by their close and accurate review of this manuscript. The manuscript has been revised according to the reviewers comments, and all changes have been highlighted for ready identification. In addition, we have addressed each of the comments from you specifically, and our responses have been outlined in a comment/response format below (see responses to the comments).

Hope the revision is satisfactory and this manuscript is now acceptable for publication in your journal.

I am looking forward to hearing from you soon.

Sincerely,

Zongfu Mao, Ph.D.

Professor and Director

Global Health Institute

School of Health Science

Wuhan University

115# Donghu Road, Wuhan City 430071,P.R.China

Responses to the reviewers’ comments

Reviewer #2

COMMENTS

This paper reports the results of a cross-sectional study on different demographic, health and socioeconomic factors related to MCI among community-dwelling older people in Wuhan, China. The sample size is adequate. Overall, the paper is well-written and the study is scientifically sound. The importance of the study is well founded in the introduction which also gives adequate background information on the significance of measuring MCI. The methods are mostly well-reported. The results are reported thoroughly and precisely. The discussion section summarizes the findings and relates them to previous knowledge on the subject. Moreover, the significance of measuring MCI and the implications of the current results are well-discussed.

I feel the paper adds relecant knowledge to the literature on MCI and related factors, especially given the specific context of China. However, I have some comments that need to be addressed before the paper could be published.

Response: We thank the reviewer for all these supportive and valuable review comments.

Comment #1) Title – the title needs to be revised, as it is now it gives the impression that the study is a methods study on measuring MCI. So please revise as to better describe the actual content of the study (i.e. correlates of MCI, or factors related to MCI etc.)

Response: We have made the change accordingly.

Comment #2) The participant flow needs to be described more thoroughly, e.g. how was the study population recruited, and through which phases was the total of 622 persons reached (e.g. how many were contacted, how many of them declined or were excluded from the study based on the different criteria, how did the sample included in the study differ from those who were excluded or did not consent to the study). I would also like to know how the face-to-face interviews were organized. Furthermore, I would like to know how the authors justify the use of a much bigger sample than needed based on the sample size calculations|.

Response: We thank the reviewer for this excellent comment. We had very strict inclusion and exclusion criteria, as we stated in the Participants section. With the permission and invaluable help of the local authorities, we had at least one community administrator in each community to help us. Because community administrators are familiar with residents, few residents refused to participate. Its household survey, the community administrator made an appointment with the residents in advance, and we conducted the interview at the appointed time. We did not track how many in total were contacted, and how many of them declined or were excluded. So, we have added more explanation to the revised manuscript. Please see the revised Participants and Limitations (the last paragraph of the discussion) sections.

Comment #3) I wonder did the authors consider conducting also adjusted logistic regression models to examine which factors maybe explained the current associations. It would be interesting to know which of the factors had the strongest association with MCI despite other contributing factors.

Response: Yes, the binary logistic regression models presented in the current study are the adjusted models. Causal inference is not implied, as the study was restricted to a cross-sectional design.

Comment #4) The discussion sections lacks strengths and limitations of the current study, please add these, especially the limitations need to be added, e.g.on the selection of the study sample.

Response: We have re-organized the related information in the revised manuscript as suggested accordingly. Please see the last paragraph of the discussion.

Comment #5) The first sentence of the results section is unclearly formulated, please correct to be more clear. This may be a language issue, I would recommend language editing throughout the manuscript as I noticed some other minor language issues as well.

Response: We have gone over the whole manuscript, and made changes and corrections according to your advice.

Comment #6) On page 5, line 189 “were less likely” is not correctly phrased based on a chi-square test, please state a fifferene and not likelihood. 

Response: We also conducted the linear trend chi-square tests, and this statement is based on the linear trend chi-square tests.

Comment #7) The result tables are quite large and therefore I would omit reporting betas, standard deviations and wald tests in the tables, these are not needed when reporting odds ratios, confidense limits and p-values.

Response: We agreed with the reviewer for the comment and have made this edit.

Comment #8) Table 3 was not clear to me, please clarify in the table captions what the types and numbers refer to.

Response: Table 3 is the matrix diagram of MCI and ADL. There are six different combinations based on the status of ADL and cognitive function of the respondents. Among which, older adults with normal cognitive and ADL functions (Type A) were selected as the “Healthy” group, while older adults with either MCI or decline in ADL functioning alone (Type B, C, D, & E), and individuals with comorbid MCI and decline in ADL functioning (Type F) were identified as “Non-complete health” group. The majority of respondents (338) were detected as Type A in “Healthy” group, while the total of “Non-complete health” groups (Type B, C, D, E & F) was 284.

We have added more context to the text in the section 3.3. Please see the revised section.

Again, we thank the reviewer for all these valuable review comments.

Round 2

Reviewer 1 Report

I would like to thank the authors for thoroughly addressing the concerns I raised on the original version of the manuscript. The authors have done a good job in the revision process. Nevertheless, there are multiple issues with the revised version that need to be addressed before this work can be accepted for publication.  

Major issues:

1)     Power planning: I appreciate the additional information provided by the authors. However, my main question remains unanswered: For what test did the authors plan their sample size? A-priori sample size estimation based on intended power is always specific to a certain statistical test, but the authors do not mention for what test they planned their sample size. Was it a chi-square test (if so: with how many df)? Was it a single regression coefficient in the logistic regression? Or was it the overall fit of the logistic regression model?

2)     P.4, l.154-156: “Multi-factor analysis was performed by establishing the binary logistic regression models to identify the most vulnerable populations who were at higher risk of MCI (Table 2) or comorbid ADL decline and MC”: I am not familiar with the term multi-factor analysis, and at the very least this term is misleading (it insinuates some sort of exploratory / confirmatory factor analysis). Please rephrase this sentence.

3)     In response to my comments in the first review, the authors have added df to the chi-square statistics in the table, but not in the main text. I think they should be reported in the main text as well.

4)     The presentation of linear qui scare tests in Table 1 is not ideal. I would suggest to report only one chi-square test per variable in Table 1.

5)     P.11, l.288-289: “One of the major findings from this study is that BMI is significantly linked with cognitive state among community-dwelling older adults”: In my opinion, this conclusion cannot be drawn from the results reported in the study. In fact, BMI was unrelated to cognitive status (see Table 1 and Table 2), it was only predictive of the combined ADL/MCI variable. This pattern of findings to me suggests that BMI is linked to ADL, but not MCI.

6)     P.12, l.317-319: “Given the convincing evidence that intervention measures are in fact effective in preventing conversion form MCI to dementia and/ or even helping these people diagnosed with MCI return to being neurologically intact are still insufficient.”:  I do not understand this sentence. Does this mean that there is convincing evidence that interventions can be effective? Or does it mean that evidence for this assumption is still lacking.

7)     I strongly suggest that the text be proofread by an English native speaker. Some sentences lack clarity (see e.g., my comment 6).

Minor issues:

8)     P.3, l.113: “There should be adequate power at all waves since”: To what waves do the authors refer here? I thought the present study was a cross-sectional study?

9)     Section Heading 3.2.: “Predictors of Factors Associated with MCI among the Elderly” seems redundant. Please consider changing to “Predictors of MCI among the Elderly”

10) P.7, l.216: Please remove “Multivariate”

11) P.8, l. 218-219: Should be one sentence: “Given that […] daily living, the current study […]”

12) P.9, l.236: “with the odds of 1.874 and 3.782”: should be “with the odds ratios of […]”

13) P.10, l. 268: Please remove “multivariate”

Author Response

Manuscript ID:

ijerph-384602 

Title:

"Correlates of Mild Cognitive Impairment of Community-Dwelling Older Adults in Wuhan, China"

Dear Reviewer #1,

We very much appreciate the thoughtful and critical feedback. Whilst we had addressed the concerns raised by the reviewers, you requested some further clarification and revisions to our manuscript, which we have completed. Please find the attached for our responses to your comments.

Again, thank you for the valuable review comments thoughout the entire process. We hope the revision is satisfactory, and this manuscript is now acceptable for publication.

Sincerely,

Zongfu Mao, Ph.D.

Professor and Director

Global Health Institute

School of Health Science

Wuhan University

115# Donghu Road, Wuhan 430071,P.R.China

Round 3

Reviewer 1 Report

I would like to thank the authors for the thorough responses. I feel that the revision has improved the manuscript substantially. I have no further concerns.